# Progress of Polymer-Based Dielectric Composites Prepared Using Fused Deposition Modeling 3D Printing

**DOI:** 10.3390/nano13192711

**Published:** 2023-10-06

**Authors:** Xueling Hu, Alix Marcelle Sansi Seukep, Velmurugan Senthooran, Lixin Wu, Lei Wang, Chen Zhang, Jianlei Wang

**Affiliations:** 1College of Chemistry, Fuzhou University, Fuzhou 350116, China; huxueling@fjirsm.ac.cn; 2CAS Haixi Industrial Technology Innovation Center in Beilun, Ningbo 315830, China; 3CAS Key Laboratory of Design and Assembly of Functional Nanostructures, Fujian Key Laboratory of Nanomaterials, Fujian Institute of Research on the Structure of Matter, Chinese Academy of Sciences, Fuzhou 350002, China; alixmarcelle@fjirsm.ac.cn (A.M.S.S.); velmurugan@fjirsm.ac.cn (V.S.); lxwu@fjirsm.ac.cn (L.W.); 4College of Chemical Engineering, Zhejiang University of Technology, Hangzhou 310032, China; sumin224@zjut.edu.cn; 5School of Materials and Chemistry Engeering, Minjiang University, Fuzhou 350108, China

**Keywords:** dielectric, polymer, composite, FDM

## Abstract

Polymer-based dielectric composites are of great importance in advanced electronic industries and energy storage because of their high dielectric constant, good processability, low weight, and low dielectric loss. FDM (Fused Deposition Modeling) is a greatly accessible additive manufacturing technology, which has a number of applications in the fabrication of RF components, but the unavoidable porosity in FDM 3D-printed materials, which affects the dielectric properties of the materials, and the difficulty of large-scale fabrication of composites by FDM limit its application scope. This study’s main focus is on how the matrix, filler, interface, and FDM 3D printing parameters influence the electrical properties of FDM-printed polymer-based dielectric composites. This review article starts with the fundamental theory of dielectrics. It is followed by a summary of the factors influencing dielectric properties in recent research developments, as well as a projection for the future development of FDM-prepared polymer-based dielectric composites. Finally, improving the comprehensive performance of dielectric composites is an important direction for future development.

## 1. Introduction

In recent decades, with the development of the electronics and energy storage industries, electronic circuits have had to develop in the direction of miniaturization, high integration, and intelligence to follow the trend of globalization [1,2], thus putting forward higher requirements for dielectric materials. Among various dielectrics, polymer dielectrics have received more and more attention in recent years because of their high breakdown strength ($E_{b}$) and low dielectric loss (DI), easy processability, and structural diversity. However, the low dielectric constant (<10) of a single polymer has limited its application [3,4,5], so the development of polymer-based dielectric composites with excellent properties such as a high dielectric constant, low dielectric loss, and high breakdown strength at the same time has become a hot research topic.

Three-dimensional printer technology, also known as additive manufacturing (AM) or rapid prototyping (RP), is considered the future of manufacturing and has the advantages of producing complex parts with low material loss, no fixturing, and high cost effectiveness compared to other manufacturing methods. FDM is one of the most widely used additive manufacturing processes [6,7]. Crum received a patent for Fused Deposition Modeling (FDM) in 1988 and founded Stratasys in 1989 [8]. The emergence of FDM has provided a new approach to how many products are designed and manufactured [9,10,11,12]. The principle diagram of the FDM process is shown in Figure 1, and the FDM process flow diagram is shown in Figure 2.

Thermoplastics such as polylactic acid (PLA) [15,16], acrylonitrile-butadiene-styrene (ABS) [17], polycarbonate (PC) [18], and polyurethane (PU) [19,20] are usually used as the matrix for printing composites using the FDM process, while PVDF-based composites, polycaprolactone (PCL)-based composites [21,22,23], and sandwich structures [24] of polymer-based composites have also been investigated. The conventional methods for preparing polymer-based dielectric composites are solvent casting, spin-coating, and hot pressing. The solvent casting method is a simple and effective method for preparing nanocomposites. Still, the drawback of this method is that the filler is not uniformly dispersed, and agglomerates and pores appear, which can weaken the dielectric properties as well as the mechanical strength of the nanocomposites. FDM can make the composites more homogeneous and have a lower void fraction, thus improving the dielectric properties. This paper first introduces the basic theory of dielectric materials. It mainly describes the factors influencing the dielectric properties of FDM-printed polymer-based dielectric composites and summarizes the research progress in recent years in the selection of the polymer matrix, the type, shape and size of filler, the interfacial interaction, the FDM printing technology, etc. Finally, it presents the challenges of high-dielectric-property polymer-based dielectric composites and their development prospects.

## 2. Dielectric Theory

Dielectric materials are also known as dielectrics, and these materials are characterized by their ability to undergo electrolysis. Electrolysis is the relative displacement of positive and negative charge centers within a molecule in the presence of an applied electric field, resulting in the formation of an electric dipole moment. A dielectric material transmits, stores, or records the action and effects of an electric field via induction rather than conduction. Dielectric materials are subject to polarization, loss, conductivity, and breakdown under the action of an applied electric field. The leading performance indicators are the dielectric constant, dielectric loss, polarization strength, breakdown electric field, etc. The magnitude of the dielectric constant of silicon dioxide (SiO_2_) ($\varepsilon_{r}$ = 3.9) is generally used as the dividing line and divided into high dielectric ($\varepsilon_{r}$ > 3.9) materials and low dielectric ($\varepsilon_{r}$ < 3.9) materials. High dielectric materials are mainly used in various electronic devices and power systems due to their excellent electric field uniformity and ultra-high energy storage capacity. In contrast, low dielectric materials are mainly used as electronic packaging materials by virtue of their ultra-low dielectric loss [6].

### 2.1. Dielectric Properties Indicators

#### 2.1.1. Dielectric Constant ($\varepsilon_{r}$)

A flat-plate capacitor usually consists of two parallel conductor plates sandwiching a dielectric material [25]. Under the applied electric field’s action, the dielectric dipoles are polarized. The positive and negative charges are shifted toward the two plates, respectively, to induce opposite charges at the plates. When the applied electric field is removed, the charges are retained on the plates, thus allowing the electrical energy to be stored efficiently. The charges *Q* and *V* stored in a flat-plate capacitor satisfy the following relationship:(1)Q=CV,
where V is the applied external voltage and the proportionality constant C is defined as the capacitance of this capacitor. C satisfies the following expression:(2)$$C=\frac{\varepsilon_{0}\varepsilon_{r}A}{d},$$
where *A* is the electrode area of the flat capacitor, d is the distance between the electrodes, and $\varepsilon_{0}$ is the vacuum dielectric constant with a value of 8.85 × 10^−12^ F/m. $\varepsilon_{r}$ is the relative dielectric constant of the material, which is the relative dielectric constant of the material without special description.

The definition of the dielectric constant can be further elaborated from the electromagnetic and microscopic point of view. The expressions are as follows:(3)$$P=D-\varepsilon_{0}E=\varepsilon_{0}\varepsilon_{r}E-\varepsilon_{0}E=\varepsilon_{0}\left(\varepsilon_{r}-1\right)E=\varepsilon_{0}\chi E\;,$$
where χ is the macroscopic polarization rate, *P* is the polarization intensity (or surface charge density), *E* is the applied field strength, and *D* is the potential shift.

As can be seen from the above equation, the dielectric properties of a material originate from the internal polarization of the material, which has the effect of resisting the applied field strength and eventually bringing the internal charge into equilibrium. Materials with high dielectric constants tend to effectively weaken the effect of the applied field strength, which is closely related to the ease of internal dipole polarization.

The dielectric constant reflects the electrodeposition ability of the dielectric material. The stronger the electrodeposition ability of the dielectric material, the higher the dielectric constant, and the more charge is stored in the capacitor per unit area. The dielectric constant reflects the electrodeposition ability of the dielectric material. The stronger the electrodeposition ability of the dielectric material, the higher the dielectric constant, and the more charge is stored in the capacitor per unit area. The dielectric constant reflects the electrodeposition ability of the dielectric material. The stronger the electrodeposition ability of the dielectric material, the higher the dielectric constant, and the more charge is stored in the capacitor per unit area.

#### 2.1.2. Dielectric Loss (tanδ)

The dielectric losses are mainly due to the relaxation of the dipoles inside the material and the migration of carriers. Under the effect of an alternating electric field, the change of the dipole or carrier migration rate in the material system cannot be synchronized with the applied electric field frequency, which leads to the negative work carried out by the applied electric field on the system at some point in time in the form of leakage current and heat dissipation, which become the main sources of dielectric loss. The expression of dielectric loss is as follows:(4)$$tan\delta =\frac{\varepsilon^{\prime \prime }}{\varepsilon^{\prime }}+\frac{\sigma }{2\pi f\varepsilon^{\prime }}\;,$$
where is the $\varepsilon^{\prime \prime }$ dielectric imaginary part and $\varepsilon^{\prime }$ is the dielectric real part, which is exactly equivalent to $\varepsilon_{r}$; σ is the electrical conductivity; and f is the frequency of electric field change.

It is evident that the dielectric loss inside the composite is closely related to the frequency of the electric field change and material conductivity [26]. In addition, the dielectric loss is also affected by the ambient temperature and humidity, both promoting an increase in polarization and a consequent increase in dielectric loss.

#### 2.1.3. Breakdown Field Strength ($E_{b}$)

The breakdown field strength ($E_{b}$) is the dielectric material in the imminent breakdown that can withstand the critical electric field strength, that is, the release of dielectric material in the electric field strength required to bind the charge, defined as the material unit thickness that can withstand the size of the voltage, reflecting the material’s high-voltage resistance and insulation capacity. When the electric field strength is below $E_{b}$, the dielectric material will undergo polarization. However, when the applied electric field exceeds $E_{b}$, the center of the positive and negative charges inside the dielectric material is shifted to a larger extent. It is very likely to become a free charge, resulting in the insulation failure of the dielectric material; the occurrence of this phenomenon is called the dielectric material breakdown [27]. For linear dielectric materials, there are
(5)$$U_{e}=0.5\varepsilon_{0}\varepsilon_{r}E^{2}\;,$$
where E is the strength of the applied electric field and $U_{e}$ is the energy storage density. From this equation, it can be seen that E has an exponential effect on $U_{e}$, which largely determines the energy storage capacity of the material.

Many factors affect the breakdown of dielectric materials, such as material composition, electrical conductivity, thermal conductivity, sample thickness, temperature, humidity, electrical stress conditions, interfaces, and surface defects. The dielectric breakdown contains different time scales, as shown in Figure 3a, with long-term influence mechanisms including oxygen, light, and moisture, while short-term influence mechanisms come from external electric fields and heat sources [27]. For polymeric materials, the temperature strongly influences the breakdown performance (Figure 3b) because the increase in temperature reduces the energy barrier for carrier migration in the material. Thus, carrier migration and material breakdown can be achieved at lower field strengths.

Therefore, the main means to improve the material $E_{b}$ include the design of polymer material microstructure and nanoparticle filling modification, the core of which is to reduce the internal defects, enhance the breakdown energy barrier, and finally achieve the preparation of high-energy storage films [28].

### 2.2. Polarization Theory

Dielectric materials under the action of an applied electric field do not have a long-range migration of carriers inside them similar to that in electric conductors, but energy storage is achieved through the polarization phenomenon inside the material. Under the action of an external electric field, a macroscopic dipole moment appears inside the dielectric along the electric field direction, and a bound charge appears on the dielectric surface, producing the polarization phenomenon, which is the main reason for the dielectric properties of dielectrics. The common dielectric polarizations include electron polarization, ion polarization, dipole polarization, and interface polarization [29]. And as shown in Figure 4, the frequency interval of the alternating electric field corresponding to different forms of polarization and the material internal response matter are different, which provides a theoretical basis and research direction for the preparation of certain energy storage devices with specific use-frequency requirements.

#### 2.2.1. Electron Polarization

Electron polarization can occur in any material. Under an external electric field, the electron cloud of each atom or ion in a molecule is displaced relative to the nucleus, causing the center of the electron cloud to deviate and produce an induced dipole moment in the direction of the electric field. This polarization, due to the relative displacement of the centers of positive and negative charges, is called electron polarization, and if the electric field is removed, the relative displacement is immediately restored and there is no loss of energy. The time required for electron polarization is the shortest, only 10^−15^~10^−13^ s.

#### 2.2.2. Ionic Polarization

When there is no electric field, the positive and negative ions of the ionic crystal are in equilibrium and electrically neutral, and the dipole moment vector sum is zero. Under the action of an external electric field, positive and negative ions are relatively displaced and polarized, and this polarization is called ion polarization. The time required for ionic polarization to occur is very short, about 10^−13^ s. Electron polarization and ionic polarization are instantaneous and elastic polarizations that do not require any energy consumption. People collectively refer to these two polarization processes as deformation polarization processes or induced polarization processes, and the polarization rate of deformation polarization is independent of the change in temperature.

#### 2.2.3. Dipolar Polarization

Dipole polarization, known as orientation polarization, generally occurs in polar molecules or polar polymers with an intrinsic dipole pitch. The polar dipole is subjected to the disordering effect of the thermal motion of the molecule, the ordering effect of the external electric field, and the intermolecular interaction, and there is a tendency to rotate in the direction of the electric field to produce polarization. The magnitude of the external electric field strength and the strength of the intermolecular forces affect the polarization, and the temperature influences the dipole polarization. The polarization is not instantaneous and takes a long time, usually 10^−9^ s. It belongs to the relaxation type of polarization, also called relaxation polarization. Dipole polarization mainly occurs in the low-frequency region of the dielectric spectrum.

#### 2.2.4. Interface Polarization

When two or more materials are compounded, the polarization phenomenon is caused by the accumulation of electrons or ions inside the material at the interface between the two phases due to the difference in dielectric properties between the materials. Interfacial polarization generally occurs in blended or filled polymer systems, which mainly occur in the lower frequency range of the dielectric spectrum (10^0^–10^2^ Hz) and only affects the medium at DC or low frequency. The interfacial polarization occurs for varying lengths of time, from a few seconds to tens of hours. The preparation of polymer-based dielectric composites should mainly consider the interfacial polarization effect.

#### 2.2.5. The Theory of Osmosis

The theory of “transmissivity” was proposed by S.R. Broadbent and J.M. Hammersely in 1957 to describe the flow of fluids in disordered porous media. The theory of transmissivity is essential for the study of filled polymer composites. The conductive filler is added to the matrix, and when the conductive filler content reaches a specific value, the conductivity and dielectric constant of the system increase sharply, and the osmosis phenomenon occurs. At this time, the filler content is called the osmosis threshold. The filler content, shape, size, and interface between the matrix resin and the filler will affect the threshold of osmosis; the lower the filler content, the farther the distance between the particles, the more difficult it is to contact; the higher the filler content, the closer the distance between the particles; when the filler content nears the threshold of osmosis, the particles can contact each other to form a channel; and too much filler will form agglomerates [31].

When the dielectric properties of polymer matrix composites are studied using the over permeability theory, the dielectric constants of the composites satisfy Equation
(6)$$\varepsilon =\varepsilon_{m}{\left(f_{c}-f\right)}^{-s}(f<f_{c}),$$
where ε and $\varepsilon_{m}$ denote the dielectric constant of the composite and the dielectric constant of the polymer matrix, respectively, f and fc denote the concentration of the conductive filler in the matrix and the transmissivity threshold, respectively, and *s* denotes the measurement constant associated with the material properties.

When f approaches $f_{c}$, ε increases abruptly, which can be explained by the fact that when the concentration of the conductive filler reaches the over-permeability threshold, the local electric field intensity increases and promotes the migration and accumulation of charge carriers at the filler–matrix interface, resulting in enhanced interfacial polarization and a sudden increase in the dielectric constant. When the conductive filler content exceeds the over-permeability threshold, i.e., when f is greater than $f_{c}$, the composite system is transformed from an insulator to a conductor. In order to obtain a high dielectric constant, the dispersion of the filler in the matrix should be maximized and made as close as possible to, but not exceeding, the osmosis threshold [32].

## 3. Factors Affecting the Dielectric Properties of Polymer Matrix Composites

### 3.1. Selection of Polymer Matrix

Dielectric polymers are characterized by low production costs, light mass, easy processing, high breakdown field strength, and a low dielectric constant (generally < 10). Their dielectric properties are closely related to the molecular structure; the greater the molecule’s polarity, the greater the dielectric constant, such as in polymers containing polar groups (e.g., EP, PVC, etc.). The dielectric constant is usually higher than those without polar groups (e.g., LDPE, BOPP, etc.), but their breakdown field strength decreases, and the dielectric loss increases compared to that. This is caused by the fact that the polar groups are more likely to be polarized under the applied electric field. However, PVDF not only has a high dielectric constant but also has a high breakdown field strength. Table 1 shows a comparison of the dielectric properties of some different polymers.

It was found that the most promising polymers that satisfy the initial screening step and are suitable for high energy density capacitor applications consist of at least one polar unit, -NH-, -CO-, and -O-, and at least one aromatic ring, -C_6_H_4_- and -C_6_H_2_S-.

### 3.2. Filler

#### 3.2.1. Types of Fillers

The fillers of polymer-based dielectric composites are mainly ceramics, conductors, semiconductors, and polymers.

##### Ceramic/Polymer Matrix Composites

Ceramic materials often have high dielectric constants, and the addition of ceramic fillers with high dielectric constants to polymers can compensate for the low dielectric constants of the polymers themselves. Common ceramic fillers include BaTiO_3_ (BT) [52], Ba_0.6_Sr_0.4_TiO_3_(BST) [53], CaCu_3_Ti_4_O_12_ (CCTO) [54,55], Ba_0.95_Ca_0.05_Zr_0.15_Ti_0.85_O_3_(BCZT) [56,57], etc. These ceramic fillers are less compatible with the polymer matrix, thus affecting the performance of the composite, so surface treatments and other methods will be commonly used to improve the compatibility between them.

Some scholars [58] added hydroxyapatite (HA) to chlorinated nitrile butadiene rubber ((Cl-NBR), and HA conferred both dielectric composites flame retardancy. Flame test (LOI) test results showed that the addition of HA particles improved Cl-NBR’s thermal stability and flame retardancy. The incorporation of HA increased the dielectric constant of Cl-NBR, which led to an increase in the dielectric loss of the Cl-NBR. The dielectric constant of the samples reached the maximum value when the HA loading was 7 phr.

Tiandong Zhang et al. [59] synthesized Ba_x_Sr_1-x_TiO_3_ powder using the sol–gel method, constructed core–shell structured Ba_x_Sr_1-x_TiO_3_@SiO_2_ (denoted as B_x_S_1-x_T@S) filler with SiO_2_ as the modifier, and prepared a series of composite films using the flow-delaying method with PMMA as the substrate, and the results showed that BST@SiO_2_/PMMA has a higher breakdown strength and polarization ability compared with BST/PMMA, and BST@SiO_2_ has a higher dielectric constant, in which the breakdown strength and polarization performance of Ba_0.6_Sr_0.4_TiO_3_@SiO_2_/PMMA nanocomposites were significantly improved at the same time. The maximum energy storage density reached was 19.6 J/cm^3^.

In addition, ceramic/polymer-based dielectric composites mainly use the iron electrodes of ceramic particles themselves; the higher the volume fraction of ceramics (mostly above 30 vol%), the higher the dielectric constant of the composites; a too high filler addition will make the mechanical properties and processability of the composites poor; the breakdown strength will also become lower; and its application is therefore limited.

##### Conductor/Polymer Matrix Composites

The conductors used to develop polymer-based dielectric composites are carbon nanotubes (CNTs) [60,61], graphene(Gr) [62], metal particles [63,64], etc. Among the metallic conductive fillers, Al, Cu, Ni [65], Ag [66], and their core–shell structures are mostly studied. Unlike ceramics, a small amount of conductor can cause a very high dielectric constant and a large dielectric loss in nanocomposites, which mainly use the percolation effect to obtain a high dielectric constant. And when the content of the conductive filler is lower than the percolation threshold, the obtained material is still an electrical insulator with a high dielectric constant, but if the content exceeds this value, the composite will be abruptly changed from an insulator to a conductor. Although a small amount of conductive filler results in an exponential increase in the dielectric constant of the system, the formation of conductive paths and the increase in leakage current will lead to a low breakdown strength and high dielectric loss of the material, according to the over diffusion theory. Breakdown strength is an important factor for dielectric materials and usually determines their reliability and practicality.

Fei Jia et al. [67] prepared high-dielectric silicone rubber (SR)-based nanocomposites with a microporous structure by incorporating graphene into silicone rubber using ultrasonic and mechanical mixing methods, and the dielectric constant increased and then decreased with increasing Gr content, reaching 18.14 (1 kHz) when the Gr content was 3 wt% and the expansion rate was 2, which was higher than that of the unexpanded sample (11.74), which is 55% higher than that of the pure sample (3–6), while the dielectric loss is less than 0.01. Hairong Li et al. [68], on the other hand, physically coated PVP layers on the reduced graphene oxide surface to obtain multilayer nanostructured rGO@PVP/PVDF composites with a high dielectric constant and low dielectric loss, and at 100 Hz, the dielectric constant of the composites reached 622 at 100 Hz, while the dielectric loss was about 0.2. The PVP surface functionalized layer played an important role in improving the dielectric constant and suppressing the dielectric loss.

##### Semiconductor/Polymer Matrix Composites

For the non-conductive ceramic filler, we usually need to introduce more than 50 wt% ferroelectric ceramic filler in the system, to significantly improve the overall dielectric properties of the material, but this also means that the composite breakdown performance and mechanical properties deteriorate. And according to the over-permeability theory, the conductive filler network lap will also lead to increased leakage current inside the material, increased material conductivity loss, and sharply decreased breakdown strength, which are also not conducive to the energy storage of the dielectric. Unlike the two functional fillers mentioned above, semiconductor materials may be more advantageous for the preparation of high-performance polymeric dielectric composites due to their controlled and intermediate conductivity between the insulator and conductor, which is used for semiconductor materials with higher conductivity. The commonly used semiconductor materials include magnesium oxide (MgO) [69,70], zinc oxide (ZnO) [71,72], titanium dioxide (TiO_2_) [73,74], and tin dioxide (SnO_2_) [75].

S. Pervaiz et al. [76] synthesized polymer-based ZnO nanocomposite flexible sheets (PB-ZnO-NCs) using PVDF as a substrate with significant improvements in structural, optical, and dielectric properties, and the dielectric measurements showed a sharp increase in the dielectric constant and a relatively low loss factor. When the ZnO nanofiller content was 50 wt%, the static value of the dielectric constant of the sample at 100 Hz was 13.88, which was 4.2 times higher than that of PVDF. The team [77] also synthesized flexible sheet polymer-based nanomaterials containing 5% ZnO and 15% and 20% SiO_2_ nanofillers using polylactic acid as the matrix, and dielectric measurements showed a sharp increase in the dielectric constant and a relatively low loss factor in the synthesis containing ZnO and SiO_2_ nanofillers. When the SiO_2_ nanofiller content was 20%, the static value of the dielectric constant of the sample at 100 Hz was 10.79, which was 3.4 times higher than that of pure PVA.

##### All-Organic Dielectric Composites

Using some special polymers as fillers into other polymers to obtain polymer composites can improve the compatibility problem between the two phases in dielectric composites and also meet the requirements of being light weight and flexibile in some fields, such as aerospace and wearable technology. Matrixes commonly used as all-organic dielectric compliant materials include PVDF and its copolymers [78,79,80,81], polyimide (PI) [82], epoxy resin (EP) [83], polymethyl methacrylate (PMMA), polyaniline [84], etc.

Qi-Kun Feng et al. [85] prepared rubber (NBR)/P(VDF-HFP) all-organic composite dielectric films using a typical solid solution casting method combined with a heat treatment process. The surface of the films prepared after heat treatment is flat, which can reduce the local electric field distortion and charge injection, and the dielectric constant and breakdown strength of the composites are significantly improved, with a dielectric constant of 10.08 and a discharge energy density of 11.3 J/cm^3^ for the NBR/P(VDF-HFP) film loaded with 2 wt% NBR at room temperature, which is higher than that of the P(VDF-HFP) film with 8.65 and 7.1 J/cm^3^ by 16.5% and 59.2%, respectively.

Polyaniline [86] is a special kind of conductive polymer material, and as a filler, it has good compatibility with the matrix and also improves the dielectric constant of the composite. Danian Liu et al. [87] designed a polyethylene glycol-polyaniline multi-crosslinked block (greater than triblock) copolymer (PEG-PANI) and prepared PVDF-based all-organic dielectric composites using the solution casting method. The results showed that the good solubility, dispersion in the matrix, and alternating block structure of the PEG-PANI/PVDF dielectric composites resulted in an excellent dielectric constant, dielectric loss, and breakdown strength. The researchers compared the dielectric properties with those of BaTiO_3_/PVDF, as shown in Figure 5. The PEG-PANI/PVDF composites had a breakdown strength of 10.74 kV/mm at 20 vol% and a dielectric constant of 86.64 (100 Hz), which were superior to those of BaTiO_3_/PVDF (9.89 kV/mm and 45.23).

#### 3.2.2. The Shape and Size of the Fillers

Within a certain range, the smaller the filler size, the better the dielectric properties and the higher the dielectric constant [88]. This may be because the smaller the filler size, the larger the specific surface area, and the larger the phase interface with the matrix, the interface gathers more active charges. The interface polarization ability is enhanced, thus increasing the dielectric constant. However, when the particle size is too small, the specific surface area is too large, which leads to the nanoparticles being very easy to agglomerate and the filler being unevenly dispersed in the composite, which will reduce the dielectric properties of the composite [89]. When the particle size of the filler is too large, defects such as porosity and interfacial weakening will appear inside the composite, which will lower the material’s dielectric constant. Mao et al. [90] investigated the effect of barium titanate crystals (BaTiO_3_) with different particle sizes on the dielectric properties of composites. The results showed that the dielectric constant of the composites with BaTiO_3_ nanoparticles added above 250 nm remained basically constant ($\varepsilon_{r}$ ≈ 65). As the BaTiO_3_ particle size decreases, the dielectric constant increases and reaches a maximum value of around 80~100 nm ($\varepsilon_{r}$ = 93). Then, the dielectric constant decreases with a further decrease in the particle size and reaches a minimum at 50 nm.

At the same filler content, BT nanorods with a high aspect ratio can effectively improve the dielectric constant of the composites. The reason may be that the filler with a high aspect ratio possesses a larger dipole moment than a spherical filler and has a stronger polarization ability, which results in a larger dielectric constant. In addition, the filler with a high aspect ratio has a smaller specific surface area and can be well dispersed in the matrix, which results in a higher dielectric constant at a low filler content [91]. Huiying Chu et al. [92] successfully prepared large aspect ratio TiO_2_/C nanofibers with different carbon contents by electrostatic spinning in situ, and a series of nanocomposite membranes were prepared using the solution casting method using P(VDF-HFP) as the matrix. The results show that the nanocomposites loaded with 10 vol% TO450 NF have the highest dielectric constant of 17.5 at 1 kHz, which is about 1.7 times higher than that of the P(VDF-HFP) matrix. This is attributed to the smaller specific surface area of nanofibers with a large aspect ratio compared to spherical fibers, the uniform dispersion and in-plane orientation of TiO_2_/C nanofibers in the polymer matrix is achieved, and the formation of microcapacitance in TiO_2_/C NFs doped with carbon can effectively increase the space charge and interfacial polarization of the nanocomposites, which leads to a significant improvement in the dielectric properties.

### 3.3. Interaction of Interfaces

The effective dielectric constant of polymer nanocomposites depends on the individual dielectric constants of the filler and the polymer matrix, as well as the interaction of various types of interfaces in the composite system. Nanoscale fillers have a high specific surface area, and the volume ratio of interfacial phases in the composite can be as high as 50–70 vol%, so the interfacial properties have a significant impact on the overall performance of the nanocomposites [93,94]. However, nanofillers with a high surface energy, van der Waals forces, or electrostatic forces are prone to agglomeration, resulting in poor dispersion in the polymer matrix, while mismatches in the relative permittivity or conductivity between the inorganic filler and the polymer matrix usually lead to inhomogeneous electric field distribution throughout the composite and result in a significant reduction in the breakdown strength of the dielectric composite. Therefore, understanding the interfacial interactions of polymer/inorganic particles is key to designing novel materials with the desired properties. Current methods to improve the interface between the filler and matrix include surface treatment, designing fillers with core–shell structure, and constructing topologies including sandwich or multilayer structures.

#### 3.3.1. Surface Treatment

Surfactants are usually used to treat the fillers. Typical surfactants include silane coupling agents, phosphate esters, and oligomers, which improve the dielectric properties of the composites by improving the interfacial compatibility between the fillers and the matrix and reducing the voids to lower the dielectric losses. However, it is essential to note that very large molecular weight surfactants may hinder the dispersion of ceramic particles, as longer chains have lower mobility and become increasingly ineffective in inhibiting agglomeration [95].

Shuning Liu et al. [96] prepared G@M/PEN nanocomposites by modifying the amine-functionalized zirconium cluster group MOF UiO-66-NH_2_ with excellent stability on the surface of graphene oxide (GO) and introducing the obtained GO@MOF (G@M) hybrid material into the PEN matrix. As shown in Figure 6, the resulting G@M/PEN dielectric composites exhibited a high dielectric constant of 7.94 and a low dielectric loss of 0.014 (1000 Hz) at low filler content (4 wt%), probably because the grafting of UiO-66-NH_2_ significantly enhanced the compatibility between the GO nanosheets and the PEN matrix, thus effectively suppressing the high dielectric loss caused by interfacial polarization, with the same GO content, G@M/PEN nanocomposites exhibited a much higher dielectric loss. These composites exhibited a much higher energy storage capacity, about 2.07 times that of conventional nanocomposites doped with graphene oxide. This further demonstrates that UiO-66-NH_2_ grown in situ on the GO surface can inhibit the formation of electric dendrites in the matrix resin of GO sheets under an electric field, thus improving the dielectric properties of the composite films.

#### 3.3.2. Core-Shell Structure

For ceramic fillers, the dispersion of ceramic fillers with core–shell structure in the matrix can improve the compatibility between the ceramic and polymer matrix [97,98] to achieve an effective combination of dielectric and mechanical properties. Conductive fillers can impart high dielectric constants to composites, but at the same time, their high dielectric losses hinder their development, and the use of insulating shells to encapsulate conductive fillers can effectively suppress dielectric losses [99,100,101,102].

Zhihui Chen et al. [103] synthesized a uniform core–shell structured Ag@C nanocables (Ag@C-NC) with a high aspect ratio (>600) using a one-step hydrothermal method and prepared PVDF nanocomposites using the solution casting method, as shown in Figure 7. The presence of a carbonaceous shell improves the interfacial adhesion between Ag@C-NC and the PVDF matrix, thus reducing the permeation threshold and dielectric loss of the composite. Near the permeation threshold ($f_{c}\;$ < 5.49 vol%), the Ag@C-NC/PVDF nanocomposite achieves both a high dielectric constant and a low dielectric loss. For example, at 1 kHz, the Ag@C-NC/PVDF nanocomposite has a dielectric constant of 295 and a dielectric loss of 0.084 when loaded with 6.45 vol% Ag@C-NC. More importantly, the prepared dielectric nanocomposites exhibited weak frequency dependence and small-scale temperature dependence.

In addition to the design of the single-shell-layer structure, Wenying Zhou et al. [104] investigated the effect of multi-shell-layer structured fillers on the dielectric and thermal properties of composites. They synthesized core@double-shell (CDS) structured Al particles by tailoring the filler interface, where a double-shell of amorphous and crystalline alumina wraps the metallic Al core. Figure 8 shows the schematic diagram of synthesizing core@single-shell or core@double-shell structured Al particles and the corresponding PVDF composites schematic diagram preparation. The experimental results show that the double-shell filler structure leads to a significant increase in the dielectric constant and a significant decrease in the dielectric loss of the corresponding composites, with better performance than the unfilled polymer and the polymer composites containing a single-shell filler, thanks to the enhancement of the interfacial polarization.

#### 3.3.3. Topological Structure

To improve the deficiency of single fillers, researchers have constructed topologies including sandwich or multilayer structures by incorporating two or more fillers into a polymer matrix at the same time or by incorporating nanoparticles into multiple polymer blends, where the additional insulating layers introduced can increase or maintain the breakdown strength while increasing the relative dielectric constant [105,106,107].

Lei Yin et al. [108] designed a PVDF-based dielectric composite sandwich structure. They combined the molten salt method and the Stöber approach to synthesize core–shell type Na_0.5_Bi_0.5_TiO_3_(NBT)@TiO_2_(TO) whiskers that greatly expanded the interface area, which is beneficial to enhance the interfacial polarization and buffer the local electric field distortion. Meanwhile, this topological multilayer structure not only gives the composite an additional non-homogeneous interface to block the electric tree, but also an insulating layer to hinder the electrode charge injection. The results show that the composite with NBT@TO whiskers in the middle layer with a mass fraction of 6 wt% has superior dielectric properties with a dielectric constant $\varepsilon_{r}$ of about 12.43, which is 51% higher than that of pure PVDF, tanδ decreases from 0.082 to 0.067 at 100 Hz, and its energy storage performance is significantly enhanced; $D_{max}$ is 13.99 μC/cm^2^, $U_{d}$ is 15.42 J/cm^3^, and η of 66.12%, which are 135, 383, and 45% higher than the pristine PVDF, respectively. This study provides a strategy for the future preparation of advanced polymer-based composites with excellent discharge energy density.

Ziyue Wang et al. [109] prepared three-phase composite films using dopamine (PDA) surface-modified potassium triazide (KNb_3_O_8_) as the filler and cis-ferroelectric (P-F) structured PMMA-P(VDF-HFP) as the substrate using the solution casting method. Figure 9 shows the schematic diagram of the paraelectric–ferroelectric (P-F) structure route and the composite film preparation process. The insulating PDA wrapped in the outermost layer of polar KNb_3_O_8_ improved the dispersion of KNb_3_O_8_ rods in the PMMA-P (VDF-HFP) matrix, enhanced the effective interfacial bonding between the filler and the polymer matrix, improved the breakdown strength ($E_{b}$) and polarization properties of the composite, and thus obtained high electrostatic energy storage. The results show that this composite film needs to be filled with only a small amount of PDA@KNb_3_O_8_ (1.0 wt%). Its dielectric constant (ε~9) and polarization ($P_{max}$~6.9) are greatly improved with a large $E_{b}$ (~600 MV m^−1^), resulting in a high energy density of 15.30 J cm^−3^ and a 65% discharge energy efficiency (Figure 10 and Figure 11). This study solves the contradiction between inorganic materials with a high dielectric constant and low breakdown strength and a low displacement polymer matrix in three-phase structured composites, which has promising applications in the field of flexible electronic capacitors.

### 3.4. FDM 3D Printing Process

Due to the simultaneous heating and cooling of FDM, pores are formed during extrusion and layer deposition [110,111]. The porous structure formed by the FDM process is the main factor affecting the dielectric properties of printed materials [112]. The presence of pores reduces the dielectric constant and breakdown strength of the material [113,114], while also affecting the mechanical properties of the material [115,116]. Therefore, optimization of FDM printing parameters can improve the comprehensive performance of dielectric composites.

#### 3.4.1. Print Speed

Previous studies by scholars [116,117] have concluded that higher print speeds affect the bonding of the extruded filament material to the previous filament. The bonding quality between neighboring filaments deteriorates, leading to the formation of more defects. In contrast, slower printing speeds provide enough time and energy for the recrystallization of the polymer chains, resulting in fewer voids. Conversely, the faster the printing speed and the shorter the forming time, the poorer the crystallization and the weaker the bonding between layers, leading to delamination.

Athanasios Goulas et al. [118] performed FDM 3D printing of ceramic-filled ABS composite wires at speeds ranging from 10 to 50 mm/s. As shown in Figure 12, no defects were observed on the sample surface when the printing speed was in the range of 10–20 mm/s. However, when the printing speed exceeded 20 mm/s, the sample printing failed, and the defects on the sample became more and more apparent with the increase in printing speed. This is because the printing speed directly affects interlayer bonding. For composite thermoplastics, where filler incorporation affects the extrudability of the material, it is important to use slower print speeds.

#### 3.4.2. Layer Thickness

In order to maintain good extrusion performance, the layer thickness should not exceed 20–80% of the nozzle diameter. Athanasios Goulas et al. printed a series of test samples with layer thicknesses ranging from 0.15 to 0.4 mm and characterized their dielectric properties. As shown in Figure 13, the thicker printed layers performed better than the thinner ones in terms of relative dielectric constant and loss. The sample with a thickness of 0.4 mm exhibited a higher relative dielectric constant ($\varepsilon_{r}$ = 9.06 ± 0.09) and lower dielectric loss (tanδ = 0.032 ± 0.003). This may be a result of the overall decrease in porosity within the samples and the increase in the amount of extruded material per given layer height.

#### 3.4.3. Filling Rate and Filling Method

The pores in FDM 3D-printed dielectric materials are the main factor that affects the dielectric properties of composites. In previous studies, researchers and scholars have found that the relative dielectric constant and dielectric loss can be tailored by controlling the fill rate of the printed material (the ratio between the printed material and the air that makes up the structure) [119,120]. The results presented in Figure 14 support this point, opening the door to the fabrication of more advanced RF devices.

Hector et al. [120] investigated the dielectric constant and dielectric anisotropy of PLA with different filling methods (honeycomb, waffle, and triangular patterns). The results revealed that PLA, which is typically an isotropic material, becomes anisotropic when a periodic pattern is introduced into its internal structure. Coupled resonator experiments indicated that the anisotropy is most pronounced in samples filled with 15% material in the honeycomb pattern. Conversely, samples treated with the waffle pattern exhibited the lowest anisotropy across all filling percentages. This discovery paves the way for the development of new microwave elements based on local variations and control of the induced medium’s anisotropy, thereby enhancing the circuit performance.

#### 3.4.4. Nozzle Temperature

During the FDM printing process, excessively high temperatures can lead to a reduction in print precision, model deformation, and other issues. Conversely, excessively low temperatures result in high material viscosity, potentially causing nozzle blockages and printing failures [121]. By appropriately increasing the nozzle temperature within the temperature range that can be printed, the material flow is better, the viscosity is lower, the newly extruded polymer molecules diffuse to the lower layer, and the interlayer adhesion is stronger. At the same time, gases are partially expelled, reducing the formation of pores and increasing the density of the material [122,123,124], as a result of which the dielectric properties of the composite are improved.

Optimization of the print temperature, print speed, fill rate, and print layer thickness can improve the overall performance of FDM-printed dielectric materials. In addition, FDM 3D printing technology has the advantage of strong designability and a customizable structure. For example, FDM is used to fabricate dielectric molds with a microchannel structure, and then, add liquid conductive filler to it, as shown in Figure 15. However, it is important to note that the FDM 3D-printed material creates a porous structure, and the printed structure should not be printed too thin to prevent leakage from forming inside the microchannel. This method was shown to be useful for the preparation of functional RF devices.

## 4. Summary and Outlook

The investigation of polymer-based dielectric composites has significantly advanced in the past few decades, but many challenges remain to be resolved. The specific enhancement of the dielectric properties of polymer-based dielectric composites requires knowledge of the parameters that affect the composites. Current research on creating and improving polymer-based composite dielectric materials has produced some promising results. However, there are still some pressing issues that need to be resolved.

In inorganic filler/polymer matrix composites, the interface between filler particles and resin matrix has not improved, and the process involved has not been comprehensively summarized. Ceramics may improve the dielectric constant of composites but have poor compatibility with the polymer matrix, decreasing breakdown strength and restricting the use of composites. The primary disadvantage of a conductive filler is that a small number of conductors can result in significant dielectric losses.

Porosity is the main factor affecting FDM 3D printing of dielectric composites. The ability to control the porosity of the printed material enables the customization of the dielectric constant and dielectric loss of the composite, opening up possibilities for the fabrication of advanced RF devices through FDM technology. However, it is important to note that while porosity manipulation offers these advantages, it also comes with certain drawbacks. The presence of porosity tends to reduce the breakdown strength of the composites and negatively impacts their mechanical properties, among other limitations. As a result, the application of FDM 3D-printed dielectric materials is somewhat constrained.

The advancement of multi-component composites, filler surface modification, and core–shell structure design are the research areas with some of the most exciting prospects.

Future research may emphasize the overall performance of materials (mechanical properties, high-temperature resistance, etc.) and focus on the dielectric properties of composites so that the materials can have a wide range of practical applications. Future developments in various disciplines, the increasingly complex electronics industry, and energy storage will all benefit from the increasing and extensive application of 3D printing technology in polymer-based dielectric materials.

## Figures and Tables

**Figure 1 nanomaterials-13-02711-f001:**
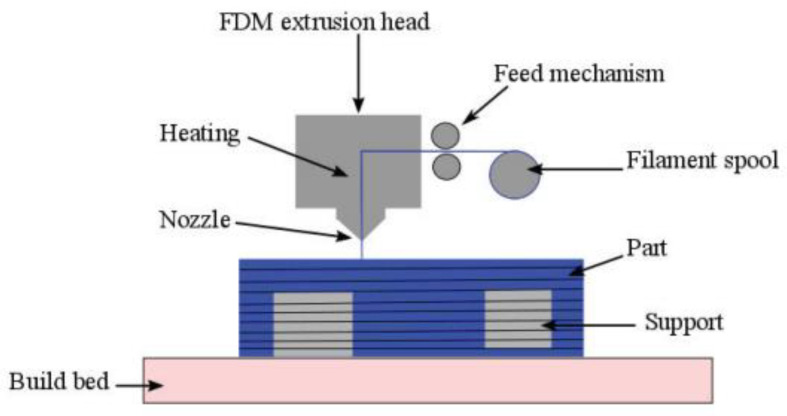
FDM process schematic [13].

**Figure 2 nanomaterials-13-02711-f002:**
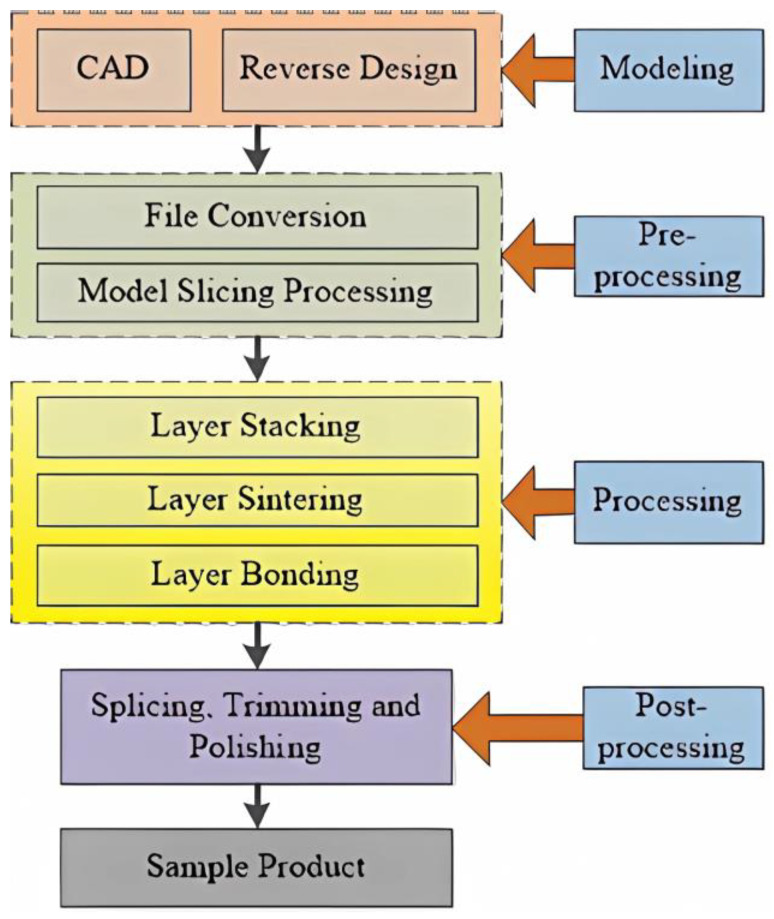
Technique flow chart of FDM [14].

**Figure 3 nanomaterials-13-02711-f003:**
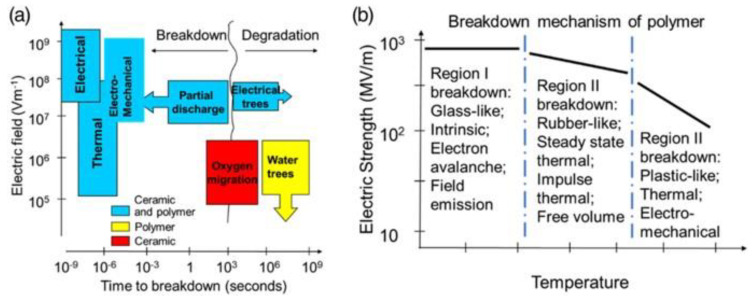
(**a**) Breakdown mechanisms of different dielectric materials; (**b**) temperature dependence of breakdown field strength [27].

**Figure 4 nanomaterials-13-02711-f004:**
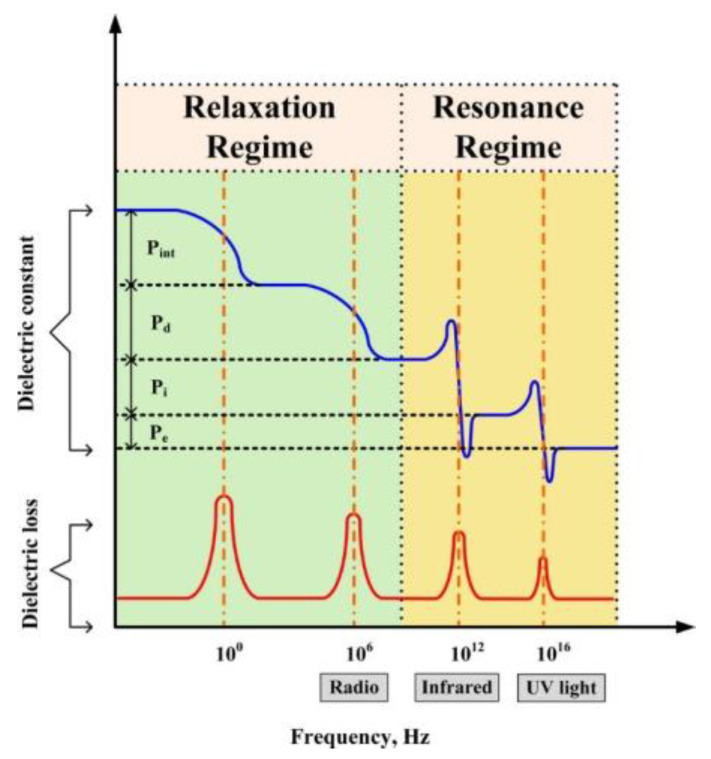
Different types of polarizations and their frequency dependence. Here, P_e_, P_i_, P_d_, and P_int_ refer to electron polarization, ion polarization, dipole polarization, and interfacial polarization. The dielectric constants and the corresponding losses are indicated by the blue and red lines, respectively. Reproduced with permission from [30]. Copyright American Chemical Society, 2016.

**Figure 5 nanomaterials-13-02711-f005:**
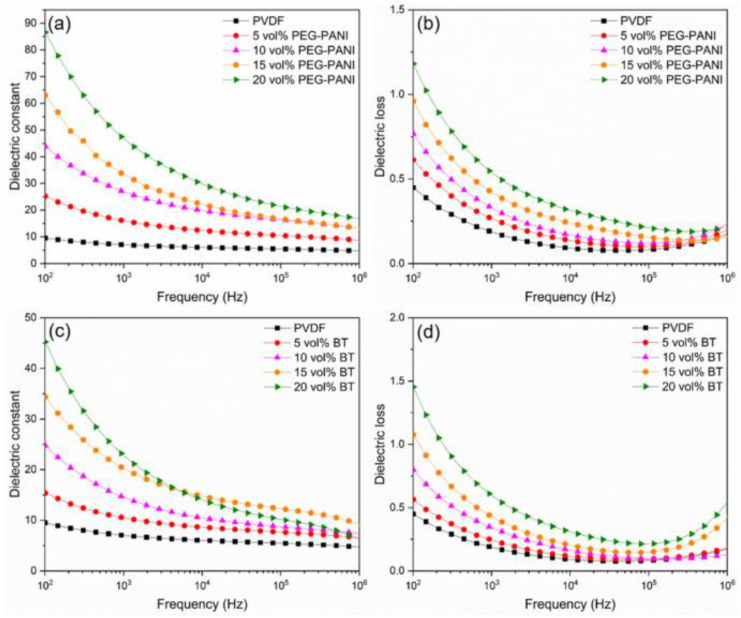
(**a**) Dielectric constant and (**b**) dielectric loss versus frequency for PEG-PANI/PVDF composites and (**c**) dielectric constant and (**d**) dielectric loss versus frequency for BT/PVDF composites at room temperature. Reproduced with permission from [87]. Copyright Elsevier, 2022.

**Figure 6 nanomaterials-13-02711-f006:**
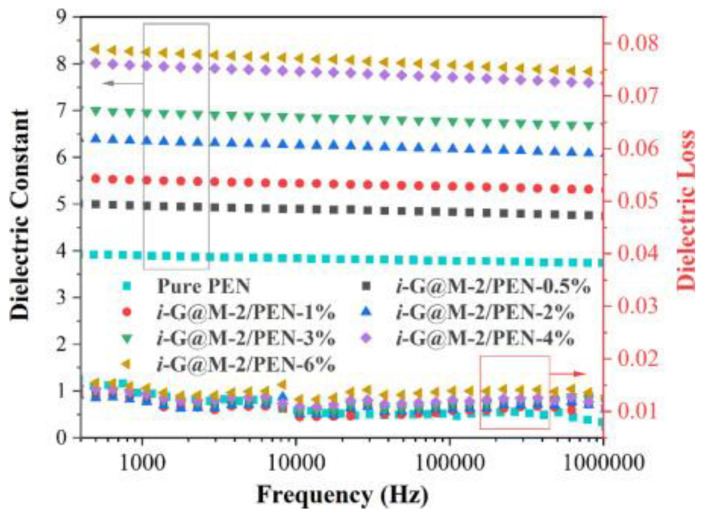
Dielectric constants and losses of i-G@M-2/PEN composites with different loadings. Reproduced with permission from [96]. Copyright Elsevier, 2023.

**Figure 7 nanomaterials-13-02711-f007:**
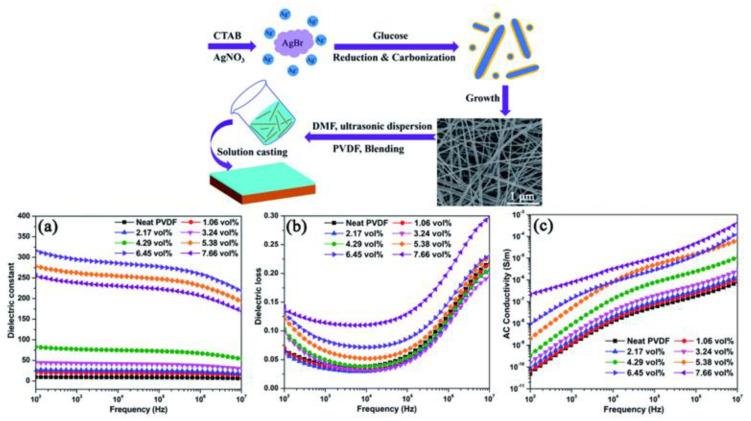
Schematic diagram of Ag@C-NC/PVDF nanocomposite preparation and dependence of (**a**) dielectric constant, (**b**) dielectric loss, and (**c**) electrical conductivity on frequency of Ag@C-NC/PVDF nanocomposite [103].

**Figure 8 nanomaterials-13-02711-f008:**
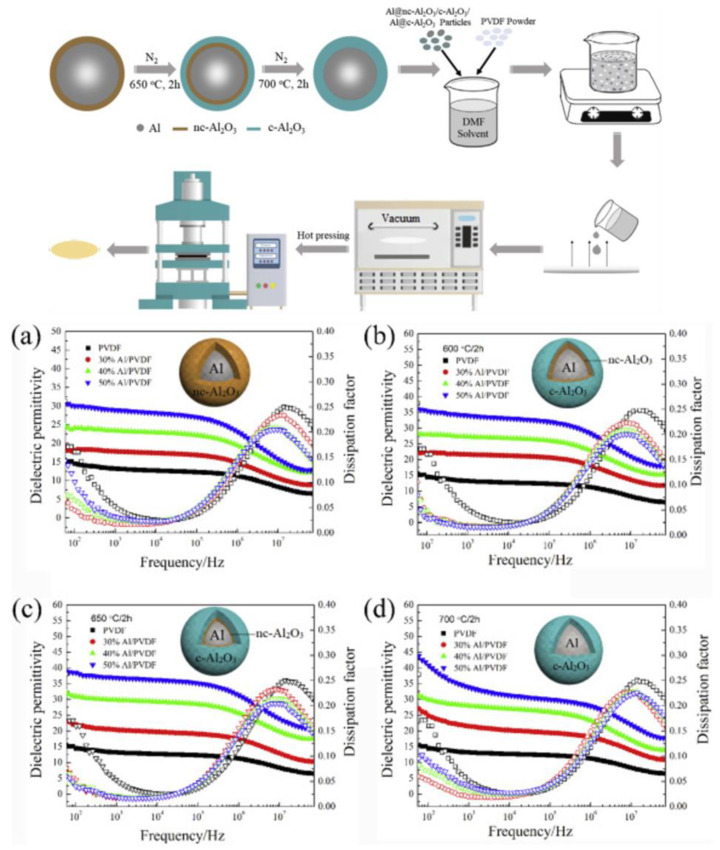
Schematic diagram of the synthesis of Al particles with core@single-shell or core@double-shell structure and the preparation of the corresponding PVDF composites and dielectric properties as a function of frequency for the unfilled polymers and the composites filled with different types of Al fillers. Filler types: (**a**) nc-Al_2_O_3_ single-shell-coated Al particles (**b**) nc-Al_2_O_3_ and c-Al_2_O_3_ double-shell-coated Al particles in order from the inside to the outside, (**c**) c-Al_2_O_3_ and nc-Al_2_O_3_ double-shell-coated Al particles in order from the inside to the outside, (**d**) c-Al_2_O_3_ single-shell-coated Al particles. Reproduced with permission from [104]. Copyright Elsevier, 2019.

**Figure 9 nanomaterials-13-02711-f009:**
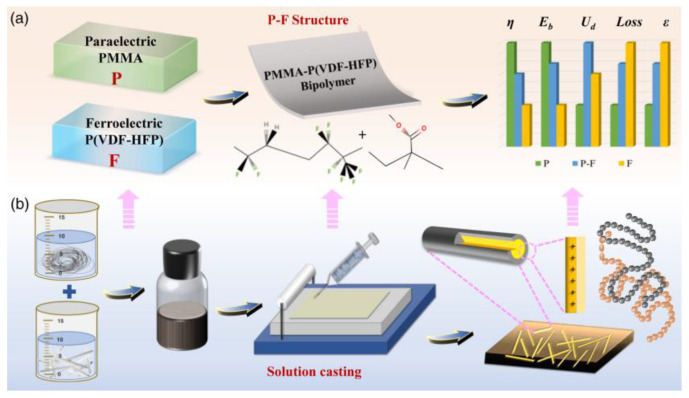
Paraelectric–ferroelectric (P-F) structure route and composite film preparation process. (**a**) P-F structure design with enhanced dielectricity and energy storage. (**b**) Monolayer composite preparation process and multicore model for PDA@KNb_3_O_8_ rod/polymer interface. Reproduced with permission from [109]. Copyright American Chemical Society, 2022.

**Figure 10 nanomaterials-13-02711-f010:**
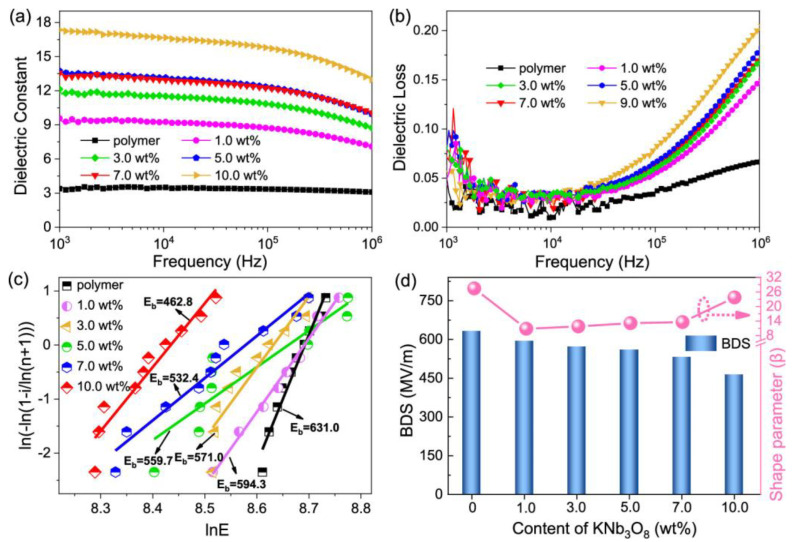
Electric and dielectric properties of the PDA@KNb_3_O_8_/PMMA-P(VDF-HFP) composite films. (**a**) Frequency-dependent dielectric constant and (**b**) dielectric loss. (**c**) Weibull distributions. (**d**) Shape parameters (β) histogram and derived breakdown strengths (BDS). Reproduced with permission from [109]. Copyright American Chemical Society, 2022.

**Figure 11 nanomaterials-13-02711-f011:**
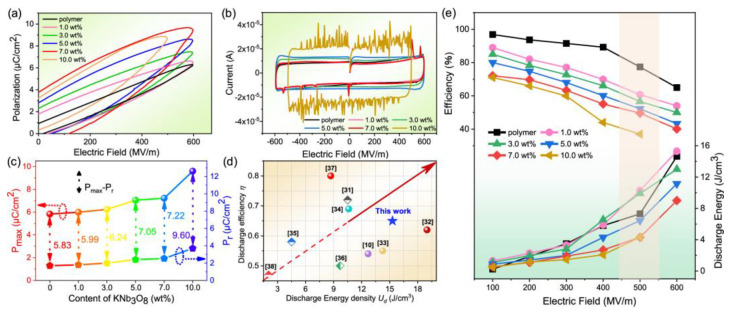
Energy storage properties of PDA@ PDA@KNb_3_O_8_/PMMA-P(VDF-HFP) composite films. (**a**) Ferroelectric unipolar P−E loops. (**b**) Current variation of different filler contents with the applied voltages. (**c**) Maximum polarization (P_max_), residual polarization (P_r_), and net polarization (P_max_ − P_r_) as a function of the PDA@KNb_3_O_8_-based fillers. (**d**) Comparison of discharge energy storage density (U_d_) and energy storage efficiency (η) among this work and some latest reports. (**e**) Discharge efficiency and energy density of the composite films. Reproduced with permission from [109]. Copyright American Chemical Society, 2022.

**Figure 12 nanomaterials-13-02711-f012:**
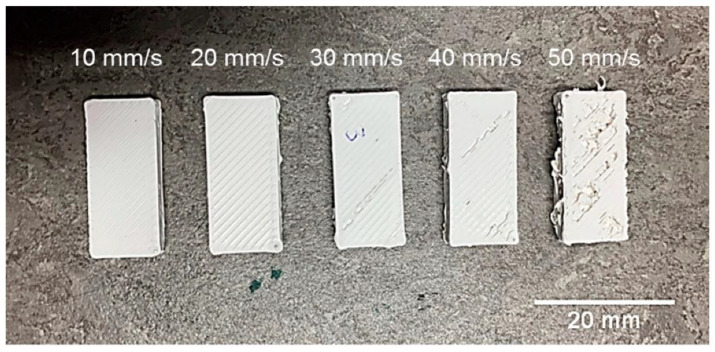
FDM 3D-printed test samples using printing speeds of 10–50 mm/s [118].

**Figure 13 nanomaterials-13-02711-f013:**
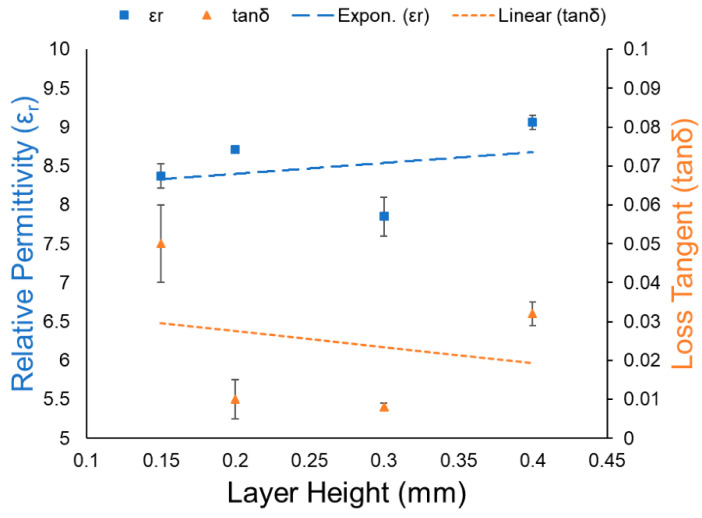
Layer height vs. relative permittivity ($\varepsilon_{r}$) and loss tangent (tanδ) [118].

**Figure 14 nanomaterials-13-02711-f014:**
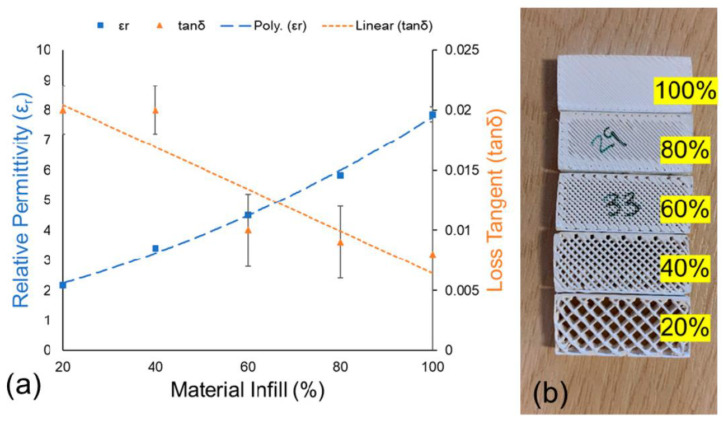
(**a**) Material infill (%) vs. relative permittivity (*ε_r_*) and loss tangent (*tanδ*); (**b**) examples of built test samples with variable material infill [118].

**Figure 15 nanomaterials-13-02711-f015:**
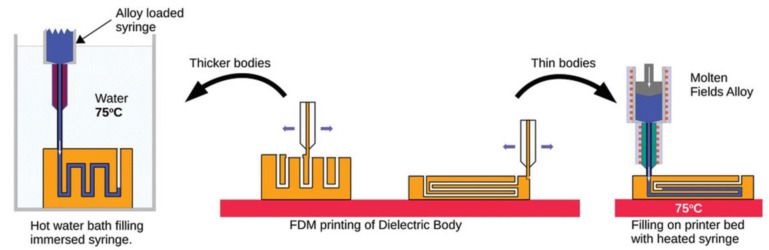
Stages in the fabrication of conductive inclusions and RF devices. (**left**) Filling of the bodies either via hot water bath injection molding of the conductors, (**center**) production of dielectric bodies with buried canals to contain the conductors, or (**right**) directly injecting molten Field’s alloy via a heated syringe into the as-printed body warmed in situ on the printer hotbed. The arrows point to separate methods of molding different bodies [125].

**Table 1 nanomaterials-13-02711-t001:** Dielectric properties of common polymer materials.

Polymers	Dielectric Constant (1 kHz)	Dielectric Loss (tanδ-1 kHz)%	Ref.
LDPE	2.3	0.3	[33,34]
PTFE	2	0.01	[35]
Polypropylene (PP)	2.2	0.02	[36,37]
Polyester (PET)	3.6	0.5	[38,39]
PMMA	4.5	5	[40,41]
Polyvinyl chloride (PVC)	3.4	1.8	[42]
Polyetheretherketone (PEEK)	4	0. 9 (100 kHz)	[43,44]
Polycarbonate (PC)	2.8	0.15	[45,46]
Epoxy	4.5	1.5	[47,48,49]
Polyethylene naphthalate (PEN)	3.2	0.15	[38,50]
Polyvinylidene-fluoride (PVDF)	12	1.8	[38,50]
Polyphenylene-sulfide (PPS)	3.0	0.03	[38,51]

## Data Availability

Not applicable.

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
