# Peer review of "Progress of Polymer-Based Dielectric Composites Prepared Using Fused Deposition Modeling 3D Printing"

_nanomaterials, 2023, doi:10.3390/nano13192711_

Round 1
Reviewer 1 Report (Previous Reviewer 4)
I am now satisfied with the revision of the manuscript.
Author Response
Thank you very much for your comment.
Reviewer 2 Report (Previous Reviewer 3)
I maintain my opinion from the previous analysis and consider that the work can be published in this form.
Author Response
Thank you very much for your comment.
Reviewer 3 Report (Previous Reviewer 2)
This manuscript has been significantly improved and now warrants publication in nanomaterials.Author Response
Thank you very much for your comment.
Reviewer 4 Report (Previous Reviewer 1)
The additional content in various places does provide the reader more detailed information about the additive process pertaining to dielectrics.

A few minor corrections have been marked on the pdf.
Author Response
Please see the attachment.

This manuscript is a resubmission of an earlier submission. The following is a list of the peer review reports and author responses from that submission.
Round 1
Reviewer 1 Report
A review paper is meant to provide a comprehensive overview of the subject matter. I strongly recommend a more balanced assessment of the capabilities of FDM, i.e. the authors should at the very least acknowledge the limitations of the process in the context of mass manufacturing. If the capacitive devices using the dielectrics are mass produced, FDM's long cycle time will in no way enable adoption of the process. Suitability for specialty devices should be discussed.
There is little discussion on the role of FDM on the morphology of the printed part that may impact it's performance.

Several writing and grammatical errors have been marked, but other similar/matching occurrences should be corrected.
Meaning is unclear in places.
Reviewer 2 Report
Manuscrypt "Progress of polymer-based dielectric composites prepared by 2 fused deposition modeling 3D printing" by Xueling Hu et al. is well and carefully written, reads well. Minor printing error line 178 is : Pc, should be Pe. I recommend printing in its current form.
Reviewer 3 Report
The topic addressed by the authors in this paper is a challenge due to its complexity. 3D printing has evolved very quickly in recent years, expanding in new directions, respectively obtaining new materials and structures becomes a necessity.
Composite dielectric materials based on a polymer, together with the proposed FDM method, are presented from different points of view. Thus, the relevant physical properties are identified for the composite (dielectric constant, losses, breakdown mechanism, types of polarization, etc.). The authors also emphasize the importance of the properties of the fillers on the properties of the composite material.
Results identified in the specialized literature for dielectric/insulating materials, conductors and semiconductors are presented, respectively the influence of dimensions, geometry, morphology, applied treatments on the properties of the composite material.
The study draws attention at the end through a series of relevant conclusions for this subject, on the problems that may arise in obtaining these materials.
I believe that the work is relevant to the chosen topic and can be accepted for publication.
Reviewer 4 Report
This review is very misleading. The title is clearly defined to review FDM 3DP of polymers/composites. The work is mostly organized in such a way it presents dielectric theory (which can be found in a physics textbook), then covers dielectric polymers and composites not prepared by FDM means (solution castings, electropinning, etc.), and finishes with a very breif paragraph on FDM materials for dielectrics. While a review on this area is valuable, this review does not capture the focus of the title at all. I did a Google Scholar search on keywords "dielectric" and "FDM/3DP" and found an abundance of published works within a 3-year periods that this review does not cite. Therefore, this review simply cannot be published in its current form. I suggest major revisions as opposed to rejecting the work because the authors can expand Section 3.4 significantly.